Glenton *et al. Implementation Science* 2018, **13**(Suppl 1):14

Implementation Science

**METHOD** 

# Applying GRADE-CERQual to qualitative evidence synthesis findings—paper 5: how to assess adequacy of data

Claire Glenton[1*], Benedicte Carlsen[2], Simon Lewin[1,3], Heather Munthe-Kaas[1], Christopher J. Colvin[4], Özge Tunçalp[5], Meghan A. Bohren[5], Jane Noyes[6], Andrew Booth[7], Ruth Garside[8], Arash Rashidian[9,10], Signe Flottorp[1] and Megan Wainwright[4]

## Abstract

**Background:** The GRADE-CERQual (Confidence in Evidence from Reviews of Qualitative research) approach has been developed by the GRADE (Grading of Recommendations Assessment, Development and Evaluation) working group. The approach has been developed to support the use of findings from qualitative evidence syntheses in decision-making, including guideline development and policy formulation.
CERQual includes four components for assessing how much confidence to place in findings from reviews of qualitative research (also referred to as qualitative evidence syntheses): (1) methodological limitations; (2) coherence; (3) adequacy of data; and (4) relevance. This paper is part of a series providing guidance on how to apply CERQual and focuses on CERQual's adequacy of data component.

**Methods:** We developed the adequacy of data component by searching the literature for definitions, gathering feedback from relevant research communities and developing consensus through project group meetings. We tested the CERQual adequacy of data component within several qualitative evidence syntheses before agreeing on the current definition and principles for application.

**Results:** When applying CERQual, we define adequacy of data as an overall determination of the degree of richness and the quantity of data supporting a review finding. In this paper, we describe the adequacy component and its rationale and offer guidance on how to assess data adequacy in the context of a review finding as part of the CERQual approach. This guidance outlines the information required to assess data adequacy, the steps that need to be taken to assess data adequacy, and examples of adequacy assessments.

**Conclusions:** This paper provides guidance for review authors and others on undertaking an assessment of adequacy in the context of the CERQual approach. We approach assessments of data adequacy in terms of the richness and quantity of the data supporting each review finding, but do not offer fixed rules regarding what constitutes sufficiently rich data or an adequate quantity of data. Instead, we recommend that this assessment is made in relation to the nature of the finding. We expect the CERQual approach, and its individual components, to develop further as our experiences with the practical implementation of the approach increase.

**Keywords:** Qualitative research, Qualitative evidence synthesis, Systematic review methodology, Research design, Methodology, Confidence, Guidance, Evidence-based practice, Data adequacy, GRADE

---

* Correspondence: claire.glenton@fhi.no
[1]Norwegian Institute of Public Health, Oslo, Norway
Full list of author information is available at the end of the article

## Background

The GRADE-CERQual (Confidence in Evidence from Reviews of Qualitative research) approach has been developed by the GRADE (Grading of Recommendations Assessment, Development and Evaluation) working group. The approach has been developed to support the use of findings from qualitative evidence syntheses in decision-making, including guideline development and policy formulation.

GRADE-CERQual (hereafter referred to as CERQual) includes four components for assessing how much confidence to place in findings from reviews of qualitative research (also referred to as qualitative evidence syntheses): (1) methodological limitations; (2) coherence; (3) adequacy of data; and (4) relevance. This paper focuses on one of these four CERQual components: data adequacy.

When carrying out a CERQual assessment, we define adequacy of data as an overall determination of the degree of richness as well as the quantity of data supporting a review finding [1]. The adequacy component in CERQual is analogous to the imprecision domain used in the GRADE approach for findings from systematic reviews of effectiveness [2].

## Aim

The aims of this paper, part of a series (Fig. 1), are to describe what we mean by adequacy of data in the context of a qualitative evidence synthesis and to give guidance on how to operationalise this component in the context of a review finding as part of the CERQual approach. This paper should be read in conjunction with the papers describing the other three CERQual components [3–5] and the paper describing how to make an overall CERQual assessment of confidence and create a Summary of Qualitative Findings table [6]. Key definitions for the series are provided in Additional file 1.

## How CERQual was developed

The initial stages of the process for developing CERQual, which started in 2010, are outlined elsewhere [1]. Since then, we have further refined the current definitions of each component and the principles for application of the overall approach using a number of methods. When developing CERQual's adequacy component, we undertook informal searches of the literature, including Google and Google Scholar, for definitions and discussion papers related to the concept of adequacy and to related concepts such as data quantity, sample size and data saturation.

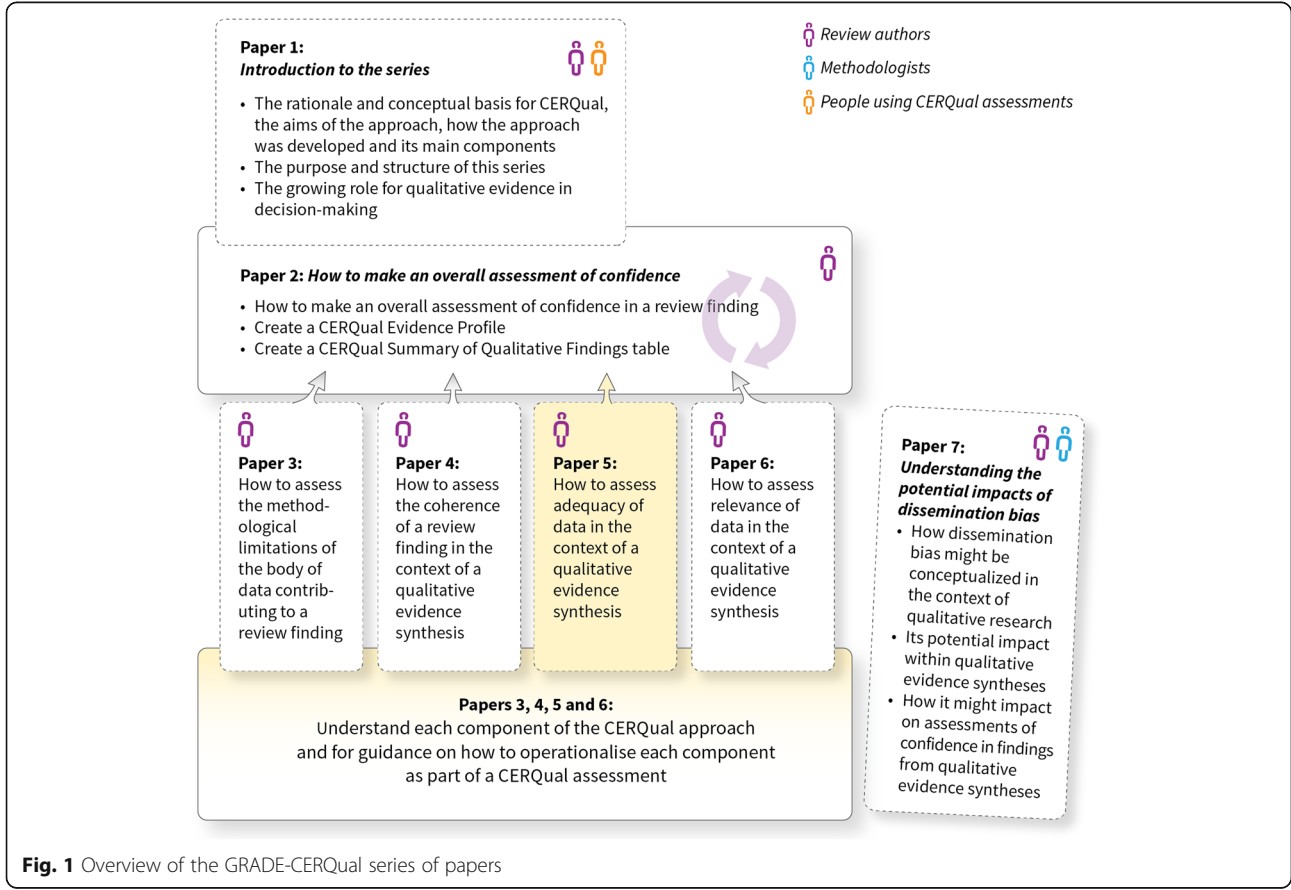

**Fig. 1** Overview of the GRADE-CERQual series of papers

We carried out similar searches for the other three components. We presented an early version of the CERQual approach in 2015 to a group of methodologists, researchers and end users with experience in qualitative research, GRADE or guideline development. We further refined the approach through training workshops, seminars and presentations during which we actively sought, collated and shared feedback; by facilitating discussions of individual CERQual components within relevant organisations; through applying the approach within diverse qualitative evidence syntheses [7–17]; and through supporting other teams in using CERQual [18, 19]. As far as possible, we used a consensus approach in these processes. We also gathered feedback from CERQual users through an online feedback form and through short individual discussions with members of the review teams. The methods used to develop CERQual are described in more detail in the first paper in this series [20].

## Assessing data adequacy
### Data adequacy in the context of qualitative research
Qualitative research methods differ from quantitative research methods in many ways, including the role that numbers play and the manner in which data adequacy is conceptualised. Typically, quantitative researchers gather a relatively limited amount of information from a large number of people and use this information to make estimates, for instance about what people do. When answering these types of questions, quantity plays an important role, and larger numbers, for instance large study populations or large numbers of event rates, generally serve as one marker of data adequacy. Qualitative researchers, on the other hand, generally study fewer people, but attempt to delve more deeply into these people's perceptions, experiences and settings in order to understand how they experience the world and why they do what they do [21]. To answer these types of questions, rich and detailed data is often required, i.e. information that is detailed enough to allow the researcher or reader to interpret the meaning and context of what is being researched [22]. For qualitative research, data richness therefore often serves as an important marker of data adequacy [22].

This emphasis on data richness does not imply, however, that numbers do not matter in qualitative research. As Sandelowski has argued, sample sizes in qualitative research "may be too small to support claims of having achieved either informational redundancy or theoretical saturation, or too large to permit the deep, case oriented analysis that is the raison-d'etre of qualitative inquiry" [23]. In other words, small numbers of study participants or observations can threaten our ability to make broad claims about a phenomenon while large numbers of participants or observations can threaten our ability to carry out a thorough qualitative analysis that would allow us to properly explore and explain a phenomenon. When determining sample size in qualitative studies, a trade-off needs to be made between conditions that require more versus fewer participants in a sample, including the aim of the study, the analytic approach and the extent to which it is based on existing knowledge and theory [24].

### Data adequacy in the context of a review finding
When assessing data adequacy in the context of a finding from a qualitative evidence synthesis, the same principles apply as for other types of qualitative research, although our focus moves from data generated from individual participants and observations to data generated from individual studies that contribute to a specific finding.

When carrying out a CERQual assessment, we define adequacy of data as an overall determination of the degree of richness as well as the quantity of data supporting a review finding.

When we talk about the *richness* of the data, we are referring to the extent to which the information that the individual study authors have provided is detailed enough to allow the review author to interpret the meaning and context of what is being researched [22]. When we refer to the *quantity* of data, we are thinking primarily of the number of studies and participants that this data comes from.

There are no fixed rules regarding what constitutes sufficiently *rich* data or an adequate *quantity* of data. Instead, any assessment of data adequacy is always relative and is judged according to "what you are expecting the data to do" [25]. In primary qualitative research, what we expect the data to do, and thereby the amount of data that we consider sufficient, will vary from study to study. While one in-depth interview could yield a rich and valid account of one person's experiences and is sufficient to document, for instance, that these experiences are within the realms of possibility, three such interviews could be sufficient to prove that people may experience the same phenomenon very differently [26]. Even higher numbers of in-depth interviews could show us how people's experiences vary and can also help us identify the experiences that many people share. The same principles apply for qualitative evidence syntheses.

When assessing data adequacy, our aim is *not* to judge whether data adequacy has been *achieved*, but to judge whether there are *grounds for concern* regarding data adequacy that are serious enough to lower our confidence in the review finding. We are likely to have concerns about data adequacy when we have concerns about the richness or the quantity of the data in relation to the claims made in the review finding.

The concept of "data saturation" is related to, although not the same as the concept of data adequacy (Table 1).

**Table 1** The relationship between data adequacy and data saturation

A related concept to data adequacy is the concept of data saturation. In primary research, "data saturation" is often used to refer to the point in data collection and analysis when "no new themes, findings, concepts or problems were evident in the data" [29]. When used in this way, the concept of data saturation is clearly different from the concept of data adequacy as the former focuses on identifying new themes while the latter concept focuses on the extent to which an individual theme or finding is adequately supported by the data. Within grounded theory, the concept of data saturation is more ambitious, however, and "relates not merely to "no new ideas coming out of the data" but to the notion of a conceptually dense theoretical account of a field of interest in which all categories are fully accounted for, the variations within them explained, and all relationships between the categories established, tested and validated for a range of settings" [25]. This second use of the concept is closer to the concept of data adequacy as both focus on the extent to which the data has allowed us to explore the topic in sufficient depth. But there are also differences between these concepts. Researchers applying the concept of data saturation in the context of primary research use this concept as an ideal or goal when collecting and analysing data, and strive to collect new data until saturation has been met. When applying the concept of data adequacy in the context of a qualitative evidence synthesis, on the other hand, researchers assess data that has already been collected, and focus on identifying concerns with this data. As the process of data saturation is potentially limitless; and determining the point at which "saturation" has happened is difficult, if not impossible [26]; the concept of data adequacy may be a more pragmatic and feasible approach.

## Guidance on how to assess data adequacy in the context of a review finding

### Step 1: collect and consider the necessary information related to adequacy (Fig. 2)

To carry out an assessment of data adequacy, you need to collect the following information for each review finding, and present it, for instance, in a table or matrix:

- An overview of the data upon which each review finding was based
- An overview of the number of studies from which this data originated, and where possible, the number of participants or observations. Information about the number of participants or observations supporting each finding may be difficult to gain from the individual studies. While most studies describe the number of participants they included in their study overall or give some indication of the extent of their observations, they may be less clear about how well represented participants are in different themes and categories. You can contact

study authors for additional information, but they may not be able to readily provide this level of detail. In these cases, this lack of information should be noted, and your assessment of data adequacy will have to be made based on the information available.

If you are using CERQual on findings from your own review, it should be straightforward to collect the information described above as you may already have constructed this type of table or matrix as part of the review process. However, if you are using CERQual to make an assessment of findings from other people's reviews, this is likely to be a time-consuming process. This is particularly the case when it comes to gaining an overview of the data upon which each review finding is based as review authors do not commonly report all of the data supporting each review finding. Unless you have access to their data extraction sheets, you will need to collect this data yourself by going back to the original references associated with each review finding. For more information on using CERQual on findings from somebody else's review, see the second paper in this series [6].

### Step 2: assess the body of data that contributes to each review finding and decide whether you have concerns about data adequacy

Once you have collected the necessary information, you can start to assess whether you have any concerns about the richness of the data supporting each review finding and the quantity of this data. (In addition to the guidance we offer here, Table 2 also presents examples of how data adequacy can be assessed for a selection of review findings. These examples illustrate how considerations of both data richness and quantity feed into the overall determination of data adequacy).

#### Assessing data richness

You are likely to have concerns about the richness of the data if it does not provide you with sufficient details to gain an understanding of the phenomenon described in

---

> Step 1: Collect and consider the necessary information related to adequacy

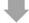

> Step 2: Assess the body of data that contributes to each review finding and decide whether you have concerns about adequacy

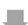

> Step 3: Make a judgment about the seriousness of your concerns and justify this judgement

**Fig. 2** Steps when assessing adequacy

**Table 2** CERQual assessments of data adequacy in the context of a review finding—examples [7, 12]

*Example 1: minor concerns*
A qualitative evidence synthesis explored factors affecting the implementation of lay or community health worker programmes for maternal and child health [12]. One of the review findings was relatively complex and explanatory in that it made claims about programme recipients' attitudes towards the lay health workers and suggested factors that appear to influence these attitudes:

*"Programme recipients were generally very positive to lay health workers. Reasons for this included the respect, kindness and concern shown by lay health workers, and their non-dogmatic approach. Recipients also appreciated the similarities they saw between themselves and the lay health workers, either because they came from the same community or because they shared similar social backgrounds."*

Twenty-five studies contributed to this finding. Ten of these studies described how recipients were generally positive to the lay health workers, but offered little or superficial information about the factors that appeared to influence these attitudes. However, nine of the studies gave more detailed and specific information about these factors. Based on an overall assessment of the richness of the data and the quantity of the data, we concluded that we had only minor concerns about data adequacy.

*Example 2: serious concerns*
Another finding from the same qualitative evidence synthesis made the following claim:

*"Recipients who lived near town and therefore had short distances to doctors preferred doctors to lay health workers"*

This finding was also relatively complex and explanatory as it suggested an association between where people live and their preferences regarding different groups of healthcare workers. However, the data upon which this finding was based offered very little information about this phenomenon, and it was not possible to properly explore or understand why doctors were preferred, and what role the distance to doctors played in people's preferences. The finding was also only based on two studies. Based on an overall assessment of the richness of the data and the quantity of the data, we concluded that we had serious concerns about data adequacy.

*Example 3: serious concerns*
A second qualitative evidence synthesis explored the mistreatment of women during childbirth in health facilities [10]. One of the review findings described a relatively unexplored phenomenon as well as making a claim that was unexpected:

*"Studies from Benin and Sierra Leone suggest that either the mother or baby may be detained in the health facility, unable to leave until they pay their hospital bills."*

Two studies contributed to this finding and the data that this finding was based on were superficial. While the finding was relatively narrow in scope, the small number of studies was of concern as the finding was unexpected and we were unsure of the extent to which studies undertaken in other settings or groups would have reported similar issues. The lack of rich data was also of concern as we were unable to properly understand this unexplored phenomenon. For instance, it was unclear from the studies whether women and babies were commonly detained, how long women and babies were detained for, and how they experienced this phenomenon. We therefore concluded that we had serious concerns about data adequacy.

*Example 4: No or very minor concerns*
A third qualitative evidence synthesis explored parents' views and experiences of communication about child vaccination [7], and included the following review finding:

*"Parents generally found the amount of vaccination information they received to be inadequate."*

Seventeen studies contributed to this finding. The data that this finding was based on were often relatively superficial. However, as the finding was a relatively simple, primarily descriptive finding, we concluded that we had no or very minor concerns about data adequacy.

*Example 5: moderate concerns*
The same qualitative evidence synthesis [7] included the following finding:

*"Parents want vaccination information resources to be available at a wider range of health services and community and online settings, for instance through schools, pharmacies, clinics and libraries."*

Only four studies contributed to this finding and both had relatively thin data, which did give us some concern. However, we judged this to be a relatively simple and descriptive finding. We therefore concluded that we had moderate concerns about data adequacy.

the review finding. This is a judgement call. However, as a rule of thumb, for review findings that are simple and primarily descriptive, relatively superficial data may be sufficient. But when a review finding is complex or explanatory, e.g. when it suggests associations between different factors, you are less likely to have confidence in that finding if it is based on data that is too superficial to allow a sufficient exploration of the phenomenon (see Fig. 3. See also Table 2 for examples). For a description of descriptive and explanatory review findings, see the paper in this series on coherence [3].

### Assessing data quantity
Alongside your assessment of data richness, you also need to consider the number of studies, participants or observations upon which the data are based. As described above, it is often difficult to gain information about the number of participants or observations that contributed to each specific finding, and you may have to focus on the number of studies.

While there is no fixed rule about what constitutes a sufficient number, you are likely to have less confidence in a review finding that is supported by data from only one or very few studies, participants or observations. This is because when only few studies or only small studies exist, or are sampled, we are less confident that studies undertaken in other settings or groups would have reported similar findings. As is the case for data richness, assessments should always be made in relation to the claims the review finding is making.

Some review findings may make claims about a limited aspect of a phenomenon or a very specific group of people or type of settings, and may need fewer studies. However, when a review finding make claims about a broad phenomenon or a large variety of people we are

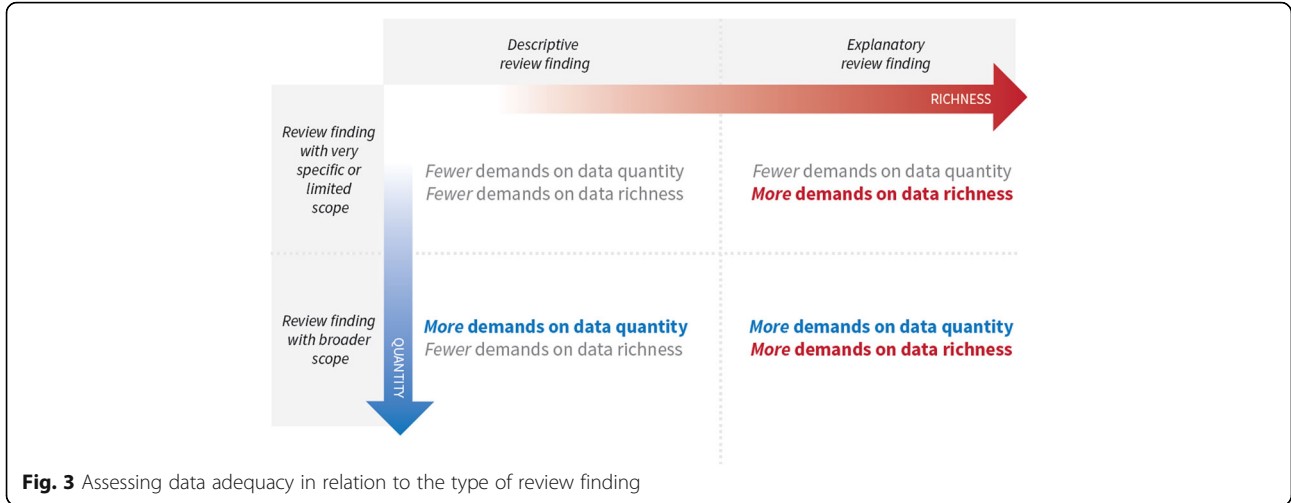

**Fig. 3** Assessing data adequacy in relation to the type of review finding

less likely to have confidence in that finding if it is based on a small number of studies (Fig. 3).

Other factors may increase your concerns about the richness or quantity of the data. Qualitative researchers aim not only to look for common attitudes and experiences, but are also interested in outliers and exceptions. However, for a review finding that makes claims about a relatively unexplored topic or a topic where people lack a widely shared language, detailed descriptions of context, intentions and meaning may be required [27]. Similarly, a review finding that makes claims that are unexpected or that challenge common knowledge may require more data than review findings that represent a widely distributed experience or domain of knowledge [28] and you may have less confidence in these findings if they are based on a small number of studies.

### Step 3: make a judgement about the seriousness of your concerns and justify this judgement

Once you have assessed data adequacy for each finding, you should categorise any concerns that you have identified as either:

- No or very minor concerns
- Minor concerns
- Moderate concerns
- Serious concerns

You should begin with the assumption that there are no concerns regarding adequacy. In practice, minor concerns will not lower your confidence in the review finding, while serious concerns will lower your confidence. Moderate concerns may lead you to consider lowering your confidence in your final assessment of all four CERQual components.

Where you have concerns about adequacy, describe these concerns in the CERQual Evidence Profile in sufficient detail to allow users of the review findings to understand the reasons for the assessments made. The Evidence Profile presents each review finding along with the assessments for each CERQual component, the overall CERQual assessment for that finding and an explanation of this overall assessment. For more information, see the second paper in this series [6].

Your assessment of data adequacy will be integrated into your overall assessment of confidence in each review finding. How to make this overall assessment of confidence is described in paper 2 [6].

### Implications when concerns regarding the adequacy of data are identified

Concerns about data adequacy may not only have implications for your confidence in a review finding, but can also point to ways of improving future research. First of all, these concerns suggest that more primary research is needed in relation to the phenomenon discussed in the review finding. The review team should also consider whether the review needs to be updated once that research becomes available.

Secondly, concerns about the adequacy of the data may indicate that certain aspects of the review question were too narrow, and that you should consider widening the scope of future updates of the review. (Any changes that are made to the scope of the review are also likely to have an impact on your assessment of the other CERQual components).

### Conclusions

Concerns about data adequacy may influence your confidence in a review finding and are therefore part of the CERQual approach. We approach assessments of data adequacy in terms of the richness and the quantity of the data supporting each review finding, but do not offer fixed rules regarding what constitutes sufficiently rich data or an adequate quantity of data. Instead, we recommend that

you make this assessment in relation to the nature of the finding. This paper aims to describe our thinking so far and help review authors and others assess data adequacy in findings from qualitative evidence syntheses. We expect the CERQual approach, and its individual components, to develop further as our experiences with the practical implementation of the approach increase.

## Open peer review

Peer review reports for this article are available in Additional file 2.

## Additional files

**Additional file 1:** Key definitions relevant to CERQual. (PDF 619 kb)

**Additional file 2:** Open Peer Review reports. (PDF 111 kb)

## Acknowledgements
Our thanks for their feedback to those who participated in the GRADE-CERQual Project Group meetings in January 2014 or June 2015 or gave comments to the paper: Elie Akl, Heather Ames, Zhenggang Bai, Rigmor Berg, Jackie Chandler, Karen Daniels, Hans de Beer, Kenny Finlayson, Signe Flottorp, Bela Ganatra, Manasee Mishra, Susan Munabi-Babigumira, Andy Oxman, Tomas Pantoja, Hector Pardo-Hernandez, Vicky Pileggi, Kent Ranson, Rebecca Rees, Holger Schünemann, Anna Selva, Elham Shakibazadeh, Birte Snilstveit, James Thomas, Hilary Thompson, Judith Thornton, Joseph D. Tucker and Josh Vogel. Thanks also to Sarah Rosenbaum for developing the figures used in this series of papers and to the members of the GRADE Working Group for their input. The guidance in this paper has been developed in collaboration and agreement with the GRADE Working Group (http://www.gradeworkinggroup.org/).

## Funding
This work, including the publication charge for this article, was supported by funding from the Alliance for Health Policy and Systems Research, WHO (www.who.int/alliance-hpsr/en). Additional funding was provided by the Department of Reproductive Health and Research, WHO (www.who.int/reproductivehealth/about_us/en/); Norad (Norwegian Agency for Development Cooperation: www.norad.no), the Research Council of Norway (www.forskningsradet.no); and the Cochrane Methods Innovation Fund. SL is supported by funding from the South African Medical Research Council (www.mrc.ac.za). The funders had no role in the study design, data collection and analysis, preparation of the manuscript or the decision to publish.

## Availability of data and materials
Additional materials are available on the GRADE-CERQual website (www.cerqual.org)
To join the CERQual Project Group and our mailing list, please visit our website: http://www.cerqual.org/contact. Developments in CERQual are also made available via our Twitter feed: @CERQualNet.

## About this supplement
This article has been published as part of *Implementation Science* Volume 13 Supplement 1, 2018: Applying GRADE-CERQual to Qualitative Evidence Synthesis Findings. The full contents of the supplement are available online at https://implementationscience.biomedcentral.com/articles/supplements/volume-13-supplement-1.

## Authors' contributions
All authors participated in the conceptual design of the CERQual approach. CG and BC wrote the first draft of the manuscript. All authors contributed to the writing of the manuscript. All authors have read and approved the manuscript.

## Ethics approval and consent to participate
Not applicable. This study did not undertake any formal data collection involving humans or animals.

## Consent for publication
Not applicable.

## Competing interests
The authors declare that they have no competing interests.

## 
## Author details
[1]Norwegian Institute of Public Health, Oslo, Norway. [2]Uni Research Rokkan Centre, Bergen, Norway. [3]Health Systems Research Unit, South African Medical Research Council, Cape Town, South Africa. [4]Division of Social and Behavioural Sciences, School of Public Health and Family Medicine, University of Cape Town, Cape Town, South Africa. [5]UNDP/UNFPA/UNICEF/WHO/World Bank Special Programme of Research, Development and Research Training in Human Reproduction, Department of Reproductive Health and Research, WHO, Geneva, Switzerland. [6]School of Social Sciences, Bangor University, Bangor, UK. [7]School of Health and Related Research (ScHARR), University of Sheffield, Sheffield, UK. [8]European Centre for Environment and Human Health, University of Exeter Medical School, Exeter, UK. [9]Department of Health Management and Economics, School of Public Health, Tehran University of Medical Sciences, Tehran, Iran. [10]Information, Evidence and Research Department, Eastern Mediterranean Regional Office, World Health Organization, Cairo, Egypt.

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
