## [Open Peer Review reports. (PDF 111 kb) · Implementation Science : IS]

Open Peer Review reports

Original Submission		
26 Oct 2016	Submitted	Original Manuscript
11 Dec 2016	Reviewed	Reviewer Report - Bridget Candy This manuscript was easy to follow and thereby easy to peer review. I have a few suggestions that may be considered by the authors. Although since I am only reading one paper out of a series, some of my suggestions may not be relevant to its final presentation within the journal. The title would be clearer if stated as 'applying to a review of qualitative research the GRADE-CERQual approach: making a CERQual assessment of adequacy of data'. (Although this may not be relevant as it will be self-explanatory as it sits within a series). Abstract: Background section. Here the authors do not state why it is important to assess adequacy of data. Although I assume this may be covered in an earlier paper in this series, but then perhaps a reminder is helpful to the reader. It may be helpful also to add that this assessment in part is similar in a certain way to GRADE evaluation for reviews of quantitative research. Results section in the last sentence, it may be helpful to the reader to know what type of/from where the information you are referring to. The background of the main text provides the aims, is there any scope to add for the sake of clarity these in the abstract? Methods: I am interested in the number of QES you used and why that number, on what criteria did you select those you used to test. The next section of the paper does not follow the format of the abstract. I guess it does not fit easily as a results section. Would the section on description and rationale for the data adequacy component fit better in the background section? This would be where such information would sit in a Cochrane review. Or would it help if this section 'of the results' has a heading like the first aim? This may make the MS flow better? To help the readers in the section on data adequacy in the context of a review finding, it would perhaps be helpful to have some examples of rich and less rich data. Tables 1 and 2 are boxes. Not certain I quite understand the difference between data saturation in primary studies and if applied in a review (page 8, line 45). Section on Guidance, would it help if the heading for this was similar to aim 2? It may be helpful if the example in Table 2 is referred to and placed earlier in the text. Also, it may be helpful to make it clear that the table includes 3 examples, perhaps by dividing in the table each example by a line?

In section in assessing data richness final sentence that data is too superficial to allow a sufficient explanation - could you details or describe further 'superficial'?

I guess using the GRADE-CERQual you could have multiple assessments (covering different findings) on adequacy of data? Or perhaps in practice you would need to focus on the few that are of primary interest?

I wonder if some aspects of points in the section on implications could also be factored into the background and as key implications in the conclusions too?

19 Jan 2017

Reviewed

Reviewer Report - GJ Melendez-Torres

This was also very clear, and a pleasure to read. I enjoyed the illuminating discussion of the difference between data saturation and data adequacy. The table documenting the scope and explanatory/descriptive aspects of a review was also welcome.

A few minor points.

On page 8, the sentence beginning with 'This second use of the concept...' was unclear: which second use was being discussed? Also, check use of data as a plural word (which it is) as opposed to data as a singular word (which it is not), e.g. later on page 8, 'data that as already been collected...'

Under Step 1, could you be more explicit about how we would collect information on the richness of the data?

Table 1 is helpful. However, I would appreciate a little bit more engagement in respect of what data richness looked like? For example, how did you find that the studies from Benin/Sierra Leone were superficial?

24 Jun 2017

Author responded

Author comments - Claire Glenton

Peer reviewer comments	
Reviewer #1:	
This manuscript was easy to follow and thereby easy to peer review.	Thank you
I have a few suggestions that may be considered by the authors. Although since I am only reading one paper out of a series, some of my suggestions may not be relevant to its final presentation within the journal.	
The title would be clearer if stated as 'applying to a review of qualitative research the GRADE-CERQual approach: making a CERQual assessment of adequacy of data'. (Although this may not be relevant as it will be self-explanatory as it sits within a series).	All the titles in the series have now been edited.

Abstract: Background section. Here the authors do not state why it is important to assess adequacy of data. Although I assume this may be covered in an earlier paper in this series, but then perhaps a reminder is helpful to the reader. It may be helpful also to add that this assessment in part is similar in a certain way to GRADE evaluation for reviews of quantitative research.	Thanks. We have added the following sentence to this paper and the other papers, both in the abstracts and the background sections. "The approach has been developed to support the use of findings from qualitative evidence syntheses in decision-making, including guideline development and policy formulation." In paper 1, we describe the relationship to other GRADE tools in more detail.
Results section in the last sentence, it may be helpful to the reader to know what type of/ or from where the information you are referring to.	The results section of the abstract has been edited as follows: "This guidance includes instructions about the information that is needed about each review finding when assessing data adequacy, the steps that need to be taken to assess data adequacy, and examples of adequacy assessments.»
The background of the main text provides the aims, is there any scope to add for the sake of clarity these in the abstract?	The abstract gives an abbreviated presentation of the aim because of word limits. The abstract states the following: "This paper is part of a series providing guidance on how to apply CERQual and focuses on CERQual's adequacy of data component.»
Methods: I am interested in the number of QES you used and why that number, on what criteria did you select those you used to test.	We have now edited the methods section of this paper to give more detail. However, due to word limitation and to avoid repetition, we have primarily described our methods in more detail in Paper 1.
The next section of the paper does not follow the format of the abstract. I guess it does not fit easily as a results section. Would the section on description and rationale for the data adequacy component fit better in the background section? This would be where such information would sit in a Cochrane review. Or would it help if this section 'of the results' has a heading like the first aim? This may make the MS flow better?	After discussion with the journal editors, we have reached agreement about the format we are using for this series of papers
To help the readers in the section on data adequacy in the context of a review finding, it would perhaps be helpful to have some examples of rich and less rich data.	As we explain in the paper, our aim is not to judge whether data adequacy (including data richness) has been achieved, but to judge whether there are grounds for concern. With this in mind, it is particularly relevant to give examples of "less rich data" rather than "rich data". In response to this and other peer review comments, we have therefore included more examples in Table 2.
Tables 1 and 2 are boxes.	The peer reviewer is correct. However, the journal only allows figures or tables, so we have named these "Tables".

Not certain I quite understand the difference between data saturation in primary studies and if applied in a review (page 8, line 45).	Thanks! This comment helped us to spot a mistake in the discussion on data saturation. Much of the text that should have been in Table 1 ended up in the main text, thereby splitting a text that should have been read in one go. We have corrected this mistake and hope that this also addresses this comment.
Section on Guidance, would it help if the heading for this was similar to aim 2?	We have now edited the aims so that they more closely reflect the heading of the section on Guidance. The aim now reads as follows: "The aim of this paper is to describe what we mean by adequacy of data in the context of a qualitative evidence synthesis, and to give guidance on how to operationalise this concept in the context of a review finding as part of the CERQual approach."
It may be helpful if the example in Table 2 is referred to and placed earlier in the text.	Table 2 is quite large and may not be able to be placed where we would like it in the text. However, we do agree that it could be referred to earlier in the guidance, and have therefore moved it further up to the beginning of the guidance.
Also, it may be helpful to make it clear that the table includes 3 examples, perhaps by dividing in the table each example by a line?	We have now added subheadings to indicate that there are different examples in Table 2.
In section in assessing data richness final sentence that data is too superficial to allow a sufficient explanation - could you details or describe further 'superficial'?	As we state in the main text, "You are likely to have concerns about the richness of the data if it does not provide you with sufficient details to gain an understanding of the phenomenon described in the review finding" . Any assessment of data as "superficial" is a judgment call, and is in relation to the phenomenon described in the review finding. To make this more tangible, we have elaborated on the third example in Table 2. We have also now referred to Table 2 at the end of this section on assessment data richness.
I guess using the GRADE-CERQual you could have multiple assessments (covering different findings) on adequacy of data? Or perhaps in practice you would need to focus on the few that are of primary interest?	In general, we would assess all findings in a synthesis, but there may be circumstances in which this is not appropriate. We have now added this discussion to Paper 2.
I wonder if some aspects of points in the section on implications could also be factored into the background and as key implications in the conclusions too?	As these are relatively short papers, we suggest that this change would be rather repetitive, and would prefer not to implement this change.
Reviewer #2:	
This was also very clear, and a pleasure to read. I enjoyed the illuminating discussion of the difference between data saturation and data adequacy. The table documenting the scope	Thank you

and explanatory/descriptive aspects of a review was also welcome.	
A few minor points: On page 8, the sentence beginning with 'This second use of the concept...' was unclear: which second use was being discussed? Also, check use of data as a plural word (which it is) as opposed to data as a singular word (which it is not), e.g. later on page 8, 'data that as already been collected...'	Thanks for spotting this. This is actually a mistake. All of the text in the paragraph starting with "This second use of the concept" should, in fact, be in Table 1. We have now moved it there.
Under Step 1, could you be more explicit about how we would collect information on the richness of the data?	We state under Step 1 that to carry out an assessment of data adequacy, you need to collect an overview of the data upon which each review finding is based. We then go on to explain, in Step 2, how to assess whether you have concerns about data richness. We have tried to make this even clearer, as follows: " You are likely to have concerns about the richness of the data if it does not provide you with sufficient details to gain an understanding of the phenomenon described in the review finding. This is a judgment call. However, as a rule of thumb, for review findings that are simple and primarily descriptive, relatively superficial data may be sufficient. But when a review finding is complex or explanatory, e.g. when it suggests associations between different factors, you are less likely to have confidence in that finding if it is based on data that is too superficial to allow a sufficient exploration of the phenomenon (See Figure 3). "
Table 1 is helpful. However, I would appreciate a little bit more engagement in respect of what data richness looked like? For example, how did you find that the studies from Benin/Sierra Leone were superficial?	We think the peer reviewer is referring to Table 2 as this is where the Benin/Sierra Leone appears. As we state in the main text, " You are likely to have concerns about the richness of the data if it does not provide you with sufficient details to gain an understanding of the phenomenon described in the review finding ". We have edited the text in Table 2 as follows: " The lack of rich data was also of concern as we were unable to properly understand this unexplored phenomenon. For instance, it was unclear from the studies whether women and babies were commonly detained, how long women and babies were detained for, and how they experiences this phenomenon. We therefore concluded that we had serious concerns about data adequacy. "

Resubmission

24 Jun 2017

Submitted

Manuscript version 2

9 Oct 2017

Author responded

Author comments - Claire Glenton

General comments from the series editor	Author responses and changes made
Thanks for providing more methodological detail in the overview and subsequent papers. There are still some areas where it would be better if you could provide further details to reflect the amount of international developmental work undertaken e.g. databases searched, timeframes, how literature reviewed etc.	We have added further detail to the overall methods description in paper 1 of the series. Specifically, we have:  - Included the years during which we ran workshops and seminars to obtain feedback on CERQual, and the numbers of workshops and presentations undertaken - Specified the period during which small group feedback sessions were run - Specified the number of CERQual users and Project Group members interviewed In the component papers (papers 3-6), we have noted that the literature searches that we undertook were informal in nature, as follows (example from paper 5): "When developing CERQual's adequacy component, we undertook informal searches of the literature, including Google and Google Scholar, for definitions and discussion papers related to the concept of adequacy and to related concepts such as data quantity, sample size and data saturation." We have also elaborated on the methods used to develop the content of paper 7 – please see below.
Ethics statements. Papers state that no humans were involved. Suggest amending to reflect consensus approach, interviews and questionnaires undertaken.	As we did not undertake formal data collection with people – all data collection was informal, in the context of training workshops, presentations and assessments of use of the approach, we have changed the ethics approval and consent to participate statements to the following: "Not applicable. This study did not undertake any formal data collection involving humans or animals."
Titles and papers could reflect paper nth of # part in a series.	We have changed all titles to the following format, as agreed earlier (example from paper 1): 'Applying GRADE-CERQual to qualitative evidence synthesis findings – paper 1 of 7: Introduction to the series'
State of the art has been removed from paper 6 but not all of the other papers in the series.	'State of the art' has been removed from all papers in the series.
The new figure outlining the process is a good addition. As a reader I would have found it easier to read papers 3-6/7 before reading paper 2.	As discussed by email with Liz Glidewell, we had a very long debate within the group about this and concluded that there is no perfect order because paper 2 (overall assessment) and papers 3-6

	(components) need to be seen together. We placed 'overall assessment' before the component papers as we felt that readers needed to understand what they were working towards before understanding each component. We feel that it would be best to keep the order as it is, but have made the following changes to assist readers: Papers 2, 3, 4, 5 and 6: We have inserted text along the lines of the following (example from paper 2 (p6): 'These component papers are closely related to this paper on making an overall CERQual assessment of confidence and creating a Summary of Qualitative Findings table. We have placed this paper before the four CERQual component papers as we think that it will be helpful for readers to understand how the component assessments will be used before discussing the details of how to apply each component.' Papers 2, 3, 4, 5 and 6: We have included in each paper an additional table that brings together all of the key definitions from each of the papers.
Do you still want to publish paper 7 as a standalone or incorporate it into the overview along with the other ongoing research?	Yes, we feel that it works best as a standalone paper.
Would the figure in the introduction outlining the process work better across all papers in the series as it contains more information than the figure just outlining the 4 and probable 5 th component?	Thanks for this very helpful suggestion which we have implemented across all of the papers.
1. Introduction	
The lack of such methods constrains the use of...suggest reframing to "methods may constrain".	Change made
"The CERQual approach is intended to be applied to well conducted syntheses." Could this be confusing to those applying the four components? Isn't CERQual designed to provide evidence of confidence in a well conducted syntheses?	We have not found this to be confusing in our interactions with users of CERQual. We feel that there would be little point in applying CERQual to a synthesis that has been poorly conducted as the findings of such a synthesis are unlikely to be reliable and the synthesis is unlikely report transparently the methods used or to include sufficient information on the primary studies to allow a CERQual assessment to be undertaken. We take the same approach in relation to GRADE for effectiveness, for the same reasons. The problem is

	sometimes colloquially called 'garbage in-garbage out'!
The section "Applying CERQual across types of qualitative data and syntheses methods". Would this be better placed after outlining how CERQual was developed?	We agree and have moved this section.
"supported other teams". Can you say any more about the scale or settings involved?	We have provided more detail as follows: "Thirdly, we applied the CERQual approach within diverse qualitative evidence syntheses in the areas of health and social care [6-8, 26-33] and also supported other teams in using CERQual by providing guidance through face-to-face or virtual training meetings and commenting on draft Summaries of Qualitative Findings tables. At least ten syntheses were supported in this way (for example, [34, 35])."
Can you provide further detail about the questionnaire and qualitative interviews?	We have now provided further detail in the text and added an additional file listing the questions covered. The revised text reads as follows: "We then gathered structured feedback from early users of CERQual through an online feedback form that was made available to all CERQual users and through short individual discussions with six members of review teams and two members of the CERQual Project Group. The questions included in the online feedback form and individual discussions are available in Additional File X."
Summarise important areas for methodological research from table 4 in text for the readers ease?	We have revised the text as follows: "Table 4 identifies several important areas for further methodological research, including how to apply CERQual in syntheses that include qualitative and quantitative data; how best to present CERQual assessments together with other kinds of evidence; ways of applying CERQual to syntheses of sources that have not used formal qualitative research procedures; and whether CERQual requires adaptation for application to more interpretive synthesis outputs, such as logic models."
2. Making an overall assessment and summary of qualitative findings	
Should the paragraph describing the four levels and rating down on p12 be moved to p10 under the 4 bulleted levels of concern?	This change has been made.
Place the text relating to variation in assessors after the text outlining who should undertake an assessment?	This change has been made.
Table 5. typo in component t missing.	This typo has been corrected.

Should you advise assessors to report how they've handled variation in levels of concern?	
3. Methodological limitations – problems design or conduct of primary studies	
Consider adding a brief description of the Evidence Profile to p12.	Ok. We have now added the following parentheses describing the evidence profile on page 12 following the sentence: “Where you have concerns about methodological limitations, describe these concerns in the CERQual Evidence Profile in sufficient detail to allow users of the review findings to understand the reasons for the assessments made (The Evidence Profile presents each review finding along with an explanation of its CERQual assessment)”
Link in text to table 2?	We have now added the following on page 9: “See Table 2 for an outline of areas where further work is needed with respect to critical appraisal tools for qualitative research.”
4. Coherence – How well finding supported by body of evidence 3500 3429	
Consider adding a brief description of the Evidence Profile to p13.	We have added a brief description of the evidence profile on page 12: “Where you have concerns about coherence, you should describe these concerns in the CERQual Evidence Profile in sufficient detail to allow users of the review findings to understand the reasons for the assessments made. The Evidence Profile presents each review finding along with the assessments for each CERQual component, the overall CERQual assessment for that finding and an explanation of this overall assessment. For more information, see the second paper in this series [19].”
5. Adequacy of data – degree of richness and quantity of data 3500 2507	

Consider contacting authors for further information as in other assessments?	We have added the following information to lines 204-205: “An overview of the number of studies from which this data originated, and where possible, the number of participants or observations. Information about the number of participants or observations supporting each finding may be difficult to gain from the individual studies. While most studies describe the number of participants they included in their study overall or give some indication of the extent of their observations, they may be less clear about how well represented participants are in different themes and categories. You can contact study authors for additional information, but they may not be able to readily provide this level of detail. In these cases, this lack of information should be noted, and your assessment of data adequacy will have to be made based on the information available.”
The sentence “For a description on descriptive and explanatory findings...” isn’t embedded.	We have moved this sentence to lines 232-233.
Consider adding a brief description of the Evidence Profile to p12.	We have added the following information to lines 277-279: The Evidence Profile presents each review finding along with the assessments for each CERQual component, the overall CERQual assessment for that finding and an explanation of this overall assessment.
6. Relevance – extent applicable to context (perspective or population, phenomenon of interest, setting) of review question 3500 3551	
I found a lot of the text more relevant to conducting a review than the CERQual assessment e.g. using theories and frameworks, how and when the review question should be developed, the pre-specification of sub-groups, strategies for identifying and selecting studies, trade-offs in searching.	Relevance is the only CERQual component that links directly to the review question. All the issues raised by the Editor need to be taken into consideration at the review design stage. We make this clear in the manuscript. See P6: ‘Relevance is the CERQual component that is anchored to the context specified in the review

	question. How the review question and objectives are expressed, how a priori subgroup analyses are specified, and how theoretical considerations inform the review design are therefore critical to making an assessment of relevance when applying CERQual.' See page 11: 'When assessing relevance, you should reflect on how the sample was located and on the underpinning principles that determined its selection....'
Word missing p13 "You should if possible, that this"	Sincere apologies, this typo was corrected previously but the corrected draft was not uploaded last time.
Is it possible to comment on how the levels of concern map onto the different threats to relevance 'partial', 'indirect' and 'unclear'?	Tables 3, 4, 5 and 6 provide visual examples. Sincere apologies, these tables may not have been uploaded in error last time.
7. Dissemination bias – selective dissemination of studies or findings 2000 2455	
Methodological details e.g. 'consulting relevant literature' and 'additional empirical work'	We have added further detail as follows: Abstract: "We developed this paper by gathering feedback from relevant research communities, searching MEDLINE and Web of Science to identify and characterize the existing literature discussing or assessing dissemination bias in qualitative research and its wider implications, developing consensus through project group meetings, and conducting an online survey of on the extent, awareness and perceptions of dissemination bias in qualitative research." Main text: "We used a pragmatic approach to develop our ideas on dissemination bias by consulting the literature on this topic, including searching MEDLINE and Web of Science to identify and characterize the existing literature discussing or

		assessing dissemination bias in qualitative research and its wider implications [3]; talking to experts in dissemination bias and qualitative evidence synthesis in a number of workshops; and developing consensus through multiple face-to-face CERQual Project Group meetings and teleconferences. We also undertook an online survey of researchers, journal editors and peer reviewers within the qualitative research domain on the extent, awareness and perceptions of dissemination bias in qualitative research [4].”
--	--	---

Resubmission 2

9 Oct 2017 Submitted Manuscript version 3

Publishing

17 Oct 2017 Editorially accepted

How does Open Peer Review work?

Open peer review is a system where authors know who the reviewers are, and the reviewers know who the authors are. If the manuscript is accepted, the named reviewer reports are published alongside the article. Pre-publication versions of the article and author comments to reviewers are available by contacting info@biomedcentral.com. All previous versions of the manuscript and all author responses to the reviewers are also available.

You can find further information about the peer review system here.